# Lead Isotopes and the Sources of Granitic Magmas: The Sveconorwegian Granite and Pegmatite Province of Southern Norway

**Nanna Rosing-Schow [1], Tom Andersen [1,2] and Axel Müller [1,3,***

[1]  Natural History Museum, University of Oslo, 0318 Oslo, Norway; nanna.rosing.schow@gmail.com (N.R.-S.); tom.andersen@nhm.uio.no (T.A.)
[2]  Department of Geology, University of Johannesburg, Auckland Park, Johannesburg 2006, South Africa
[3]  Natural History Museum of London, Cromwell Road, London SW7 5BD, UK
[*]  Correspondence: a.b.muller@nhm.uio.no

**Abstract:** Lead isotope analyses of K-feldspar from late Sveconorwegian (900–1000 Ma) granitic pegmatites and A-type, ferroan granitic intrusions in four different areas of southern Norway analyzed by laser-ablation inductively coupled plasma source mass spectrometry (LA-ICPMS) give compositions in the range $^{206}Pb/^{204}Pb$ = 16.637 to 17.555, $^{207}Pb/^{204}Pb$ = 15.445 to 15.534, $^{208}Pb/^{204}Pb$ = 36.317 to 37.459. These compositions broadly overlap with the initial compositions estimated from previously published solution TIMS whole-rock and feldspar Pb isotope analyses of late Sveconorwegian granitic plutons across the region, suggesting that magmas forming A-type granite plutons and granitic pegmatites have been derived from broadly similar source rocks, i.e., from a continental crust that initially formed in Palaeoproterozoic time (ca. 2.10–1.86 Ga), and subsequently underwent intracrustal partial melting, differentiation and rejuvenation via mafic underplating in Mesoproterozoic time.

**Keywords:** granites; pegmatites; Pb isotopes; source modelling; Sveconorwegian orogen; southern Norway

## 1. Introduction

Granites and granitic pegmatites have near-minimum melt compositions in both simple experimental systems such as albite—K-feldspar—quartz and in more complex natural rock systems e.g., [1]. Whereas residual melts of granitic composition may form by fractionation of mafic, mantle-derived magmas e.g., [2,3], most granites are likely to have formed via anatexis within the crust [4] or contain a significant fraction of melts derived from crustal protoliths [5]. Melting of a wide range of crustal protolith compositions may give rise to melts that are broadly granitic in composition [1,6]. Because granitic rocks exposed at the present continental surface carry information on the nature and composition of their sources, radiogenic isotope data on anatectic granites can be used to characterize source components in the unexposed crust, and to constrain their distribution and the evolution of the crust over time [7].

Lead isotopes in rocks formed from anatectic melts, and the K-feldspar in such rocks, are potential tracers of the U-Th-Pb characteristics of the crustal sources e.g., [8]. One of the strengths of the U-Th-Pb isotope system is the existence of three related decay series producing radiogenic Pb isotopes: $^{238}U \rightarrow {}^{206}Pb$, $^{235}U \rightarrow {}^{207}Pb$ and $^{232}Th \rightarrow {}^{208}Pb$. These decay series support each other and can give more information than single parent-daughter radiogenic isotope systems. Introduction to the interpretation of Pb isotope data in crustal rocks are given in standard textbooks e.g., [9], a formal development can be found in [10], and a more recent review of the method, including all the necessary mathematical relationships in [11].

K-feldspar incorporates elevated trace concentrations of Pb, while excluding U and Th. The Pb isotope composition of K-feldspar in an igneous rock thus retains a memory of the Pb isotopic composition of the melt from which it crystallized, at the time of formation. Since this feature is inherited from the source of the melt, Pb-isotope data on igneous K-feldspars can be used to identify the source(s) contributing material to the magma. Pb isotope data have proved to be a useful tool to study the genesis of granites and granitic pegmatites e.g., [12–14].

The focus area of the present study in southern Norway is shown in Figure 1. This region is host to several suites of late Mesoproterozoic—early Neoproterozoic granitic intrusions, as well as more than 5000 granitic pegmatites comprising several pegmatite fields [5,15,16]. Previous radiogenic isotope studies on the granite plutons indicate that granitic melts have formed by anatexis of sources varying from purely crustal [17] through mixtures of mantle- and crustal-derived components [18] to dominantly mantle derived [19,20].

Granitic pegmatites range in composition from essentially pure quartz-feldspar assemblages representing simple minimum melts, to systems that are highly enriched in trace elements such as Nb, Y, Sc, REE etc. and hosting exotic minerals [21]. Pegmatites enriched in incompatible trace elements can in principle be generated by differentiation from a granitic melt [21–23], or they can be low-volume melts formed by crustal anatexis [16,24–27]. Some pegmatite fields in the Precambrian of southern Norway are either not associated with granitic plutons exposed at the surface or associated with granites of significantly different ages [16,28–30]. The lack of contemporary relationships suggests that such pegmatites cannot be related to the exposed granites by a magmatic differentiation process, and that additional petrogenetic processes may therefore be at work in these pegmatite fields. One such process that has been suggested is local anatexis of mafic (amphibolite) protoliths [16,30,31].

Neither geochronological data nor Pb isotope data constitute sufficient evidence to confirm or disprove a petrogenetic model. However, since the Pb isotope composition of igneous K-feldspar provides information on the nature of the source of magma, and the Pb isotope characteristics of many of the granites in the region are relatively well known [5,18], Pb isotope data can help deciding if the pegmatite magmas have tapped the same lower crustal reservoirs as the A-type granites in the region, or if source rocks with other characteristics (i.e., mafic protoliths of mantle-derived origin) are dominant. In this study, we present Pb isotope data on K-feldspar analyzed by laser ablation inductively coupled plasma source mass spectrometry (LA-ICP-MS) to characterize the source(s) of granitic pegmatites from the Froland, Evje-Iveland, Tørdal, Flesberg and Østfold pegmatite fields in south Norway. We have sampled K-feldspar from the pegmatites and adjacent granites (Table 1) to compare their isotopic signatures and to identify potential crustal and mantle sources. This is compared to published isotope data that exist for the region.

**Table 1.** Sample list of investigated K-feldspars from Sveconorwegian granites and pegmatites.

| Sample | Locality | Pegmatite Field | Mineral | Color Variant | UTM Zone | UTM E | UTM N |
| --- | --- | --- | --- | --- | --- | --- | --- |
| 2109402 | Lille Kleivmyr | Froland | Kfs | Beige | 32 V | 468400 | 6495347 |
| 2009407 | Sønnristjern | Froland | Kfs | Pink | 32 V | 466795 | 6494807 |
| 26051712 | Herrefoss granite | Froland | Kfs | Pink | 32 V | 464369 | 6474742 |
| 59295 | Solås | Evje-Iveland | Kfs | Amazonite | 32 V | 437070 | 6483798 |
| 09070819 | Solås | Evje-Iveland | Kfs | Beige | 32 V | 437070 | 6483798 |
| 12070802 | Landsverk | Evje-Iveland | Kfs | Pink/beige | 32 V | 433449 | 6495851 |
| 07070802 | Steli | Evje-Iveland | Kfs | White | 32 V | 434427 | 6484276 |
| 22051701 | Høvringsvatnet granite | Evje-Iveland | Kfs | Pink | 32 V | 439111 | 6501715 |
| 14101602 | Herrebøkasa | Østfold | Kfs | Beige | 32 V | 641967 | 6551075 |
| 14101610 | Idefjord granite | Østfold | Kfs | Pink | 32 V | 641832 | 6550665 |
| 07061607 | Kleppe quarry | Tørdal | Kfs | Beige | 32 V | 484892 | 6557629 |
| 05061610 | Svåheii 3 | Tørdal | Kfs | Beige | 32 V | 481515 | 6557011 |
| 23091503 | Upper Høydalen | Tørdal | Kfs | Amazonite | 32 V | 486274 | 6560473 |
| 23091502 | Upper Høydalen | Tørdal | Kfs | White | 32 V | 486274 | 6560473 |
| 05061621 | Pegmatitic granite | Tørdal | Kfs | Pink | 32 V | 482683 | 6557045 |
| 20091501 | Tørdal granite (Skjeggefoss quarry) | Tørdal | Kfs | Pink | 32 V | 489004 | 6558103 |



## 2. Geological Setting

The rocks investigated in this study are situated within the Southwest Scandinavian Domain (SSD) of Fennoscandia [32], which comprises South Norway southeast of the Caledonian nappes, and adjacent parts of Sweden west of the Protogine Zone (Figure 1). Several important events generating or remobilizing continental crust have been identified in central and southwestern Fennoscandia: 1.9–1.75 Ga Svecofennian orogeny, emplacement of the mainly granitic Transscandinavian Igneous Belt (1.85–1.65 Ga, [33]), 1.75–1.55 Ga "Gothian orogeny" and 1.1–0.9 Ga Sveconorwegian orogeny [32,34].

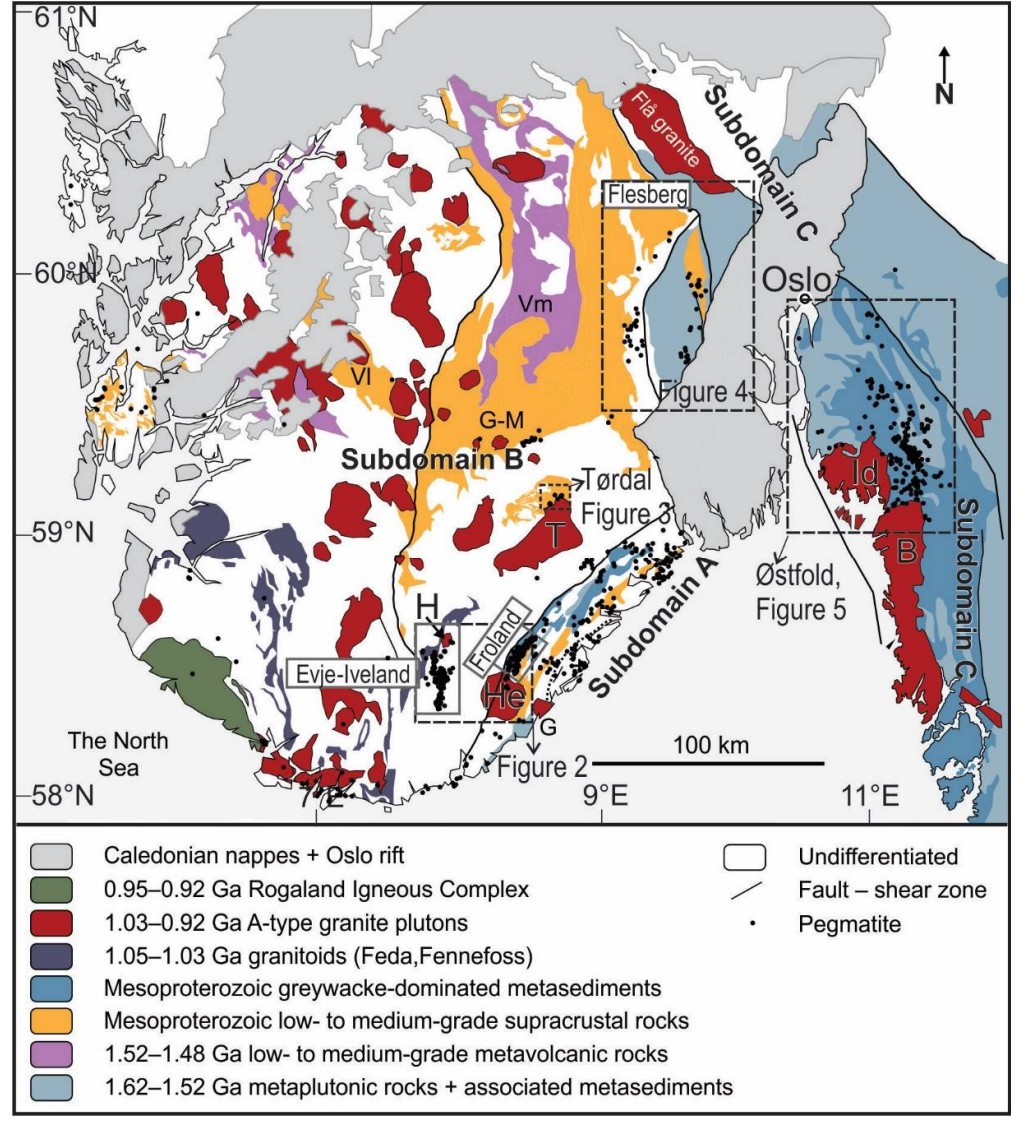

**Figure 1.** Geological sketch map of the Sveconorwegian orogen. The position of the major Meso-proterozoic mafic magmatism (Vl: Valldal (1.26 Ga), G-M: Gjuve-Morgedal (1.16 Ga), Vm: Vemork (1.5 Ga)) and the Opx-in isograd related to regional Sveconorwegian metamorphism is shown. The A-type granites studied are marked with letters: T = Tørdal-Treungen granite, H = Høvringsvatnet, He = Herefoss granite, Id = Iddefjord granite and B = Bohus granite. Modified from [35,36]. Sub-domain A corresponds to the Bamble Sector of [37], Bamble–Lillesand block of [38], and Bamble lithotectonic unit of [39]. Subdomain B comprises the Telemark and Rogaland-Vest Agder sectors of [37]; Telemark and Hardangervidda–Rogaland blocks of [38] or the Telemark lithotectonic unit of [39]. Subdomain C corresponds to the Kongsberg–Marstrand block of [38], combining the unit variously identified as the Western Segment [40], 'Idefjord Terrane' [41] or Idefjorden lithotectonic unit [39] with the Kongsberg Sector of [37] or Kongsberg lithotectonic unit of [39].

In southern Norway, no rocks older than the late Paleoproterozoic have so far been identified, and in the areas of interest in this study, the oldest exposed rocks are of early Mesooproterozoic age [20,35,42–45]. However, late Mesoproterozoic—early Neoproterozoic granites from across southern Norway show Sr, Nd, Hf and Pb isotopic evidence of having been derived from source rocks with a crustal history back to the late Palaeoproterozoic [5,44]. This source of melts does not have a classical "depleted lower crust" isotopic and trace element signature [5,44], and can be, at least tentatively, correlated with the 2.10–1.86 Ga Svecofennian juvenile crust that was the source of late Palaeoproterozoic granitioids of the Transscandinavian Igneous Belt [33,46].

The SSD is composed of several sub-domains, the definition and naming and significance of which remain controversial e.g., [35,38,39,41]. The definition and naming is not important for the present study, and we therefore subdivide the region of interest into three subdomains denoted *A*, *B* and *C*, as shown in Figure 1.

*Subdomain A* consists of metasedimentary gneisses and quartzites, associated with metaigneous quartzofeldspatic gneisses (including 1.57–1.52 Ga tonalitic to granodioritic bodies), and metagabbroic rocks [37,43,47]. Medium-pressure metamorphism to the amphibolite facies and locally to the granulite facies [48] has been dated to 1140–1080 Ma [49].

*Subdomain B* consists mainly of amphibolite-facies gneisses, including migmatites with supracrustal protoliths, granitic gneisses, augen gneisses and low grade ca. 1500 Ma and ca. 1100 Ma supracrustal formations known as the Telemark Supergroup [37]. The Mesoproterozoic units underwent extensive metamorphism in the late Mesoproterozoic (i.e., Sveconorwegian time) with metamorphic grade increasing from amphibolite facies in the east to granulite facies in the west [35]. Abundant granitoids, and felsic and mafic volcanic rocks formed around 1.5 Ga represent the first recorded magmatic event in the subdomain [20,50,51]. This magmatism includes the mainly felsic volcanic rocks of the Rjukan Group in Telemark and tonalitic intrusions in the Setesdal area [20,52]. The presence of inherited, Palaeoproterozoic zircons in Sveconorwegian granites, and the general geochemistry and radiogenic isotope systematics of late Sveconorwegian granites across the subdomain suggest the presence of older rocks in the unexposed crust that are indistinguishable from late Palaeoproterozoic granitoids of the Transscandinavian Igneous Belt [5,44]. Mafic magmatic events occurred at 1.28–1.26 Ga [53,54] and 1.17–1.14 Ga [55–57], and felsic magmatism at 1.21–1.20 Ga [19,20,58], 1.05–1.03 Ga [59], followed by the post-orogenic, A-type granitic magmatism at 1.0 Ga to 0.9 Ga [15,39,45] and intrusion of anorthosite and related rocks of the 950–920 Ma Rogaland Igneous Complex [43,59–62]. The rocks of Subdomain A have been overthrust Subdomain B along a SE-dipping, ductile shear-zone [37] that has been dated to 1070 Ma [63]. The contact relationships between the two subdomains have later been modified by a Phanerozoic (post-Silurian, most probably Permian) oblique brittle fault known as the Porsgrunn–Kristiansand Fault [64,65].

*Subdomain C* is built up of late Palaeoproterozoic to early Mesoproterozoic calc-alkaline metaigneous gneisses with a primitive continental margin arc signature, metasedimentary rocks (quarzites, metapelites, metagreywackes), penetrated by 1.62–1.56 Ga tonalitic to granodioritic intrusions [66]. Despite of the primitive geochemical signature, inherited zircons in calc-alkaline intrusive rocks indicate the presence in the early Mesoproterozoic of a Palaeoproterozoic substrate indistinguishable from the Transscandinavian Igneous Belt [66]. Subdomain C was cut by the Oslo Graben in late Palaeozoic.

*Sveconorwegian A-Type Granites and Pegmatites*

Post-orogenic, ferran and potassic A-type hornblende-biotite-bearing granites [67] intrude all the sectors of the Sveconorwegian orogen. They range in age from approximately 1030 Ma to 920 Ma [15,39,51]. Furthermore, all sectors are intruded by pegmatites representing the last recorded magmatic event in the orogen. The A-type granite suite can be divided into three groups based on comprehensive Sr, Nd and Pb isotope studies [5]: Group 1 granites ("normal Sr concentration granites") have more than 150 ppm Sr, $^{87}Rb/^{86}Sr < 5$, $^{87}Sr/^{86}Sr_{0.93Ga} < 0.710$ and $\varepsilon_{Nd} < 0$; Group 2 granites have less than 150 ppm

Sr, $^{87}Rb/^{86}Sr > 5$, $^{87}Sr/^{86}Sr_{0.93Ga} > 0.710$ and $\varepsilon_{Nd} < 0$; Group 3 comprises only one intrusion with low $^{87}Sr/^{86}Sr_{0.93Ga} < 0.710$ and $\varepsilon_{Nd} > 0$, representing a relatively uncontaminated mantle component. The Group 1 granites are the most abundant of the three groups [5,44]. The Herefoss, the Høvringsvatnet and the Iddefjord granite have Group 1 granite radiogenic isotope signature [5]. For the Tørdal-Treungen granite, radiogenic isotope data suggest a mixed character, with one of the analysed samples (from Treungen) belonging in Group 1, and one (from Tørdal) in Group 2 [5]. Nd, Sr and Pb isotopes of the suite suggest that the source of magma contains a mix between a depleted mantle component and two or more distinct crustal components [5,44], one of which comprises 1.6–1.9 Ga rocks equivalent of the Transscandinavian Igneous Belt of central Fennoscandia, the other ca. 1500 Ma rocks related to the Rjukan Group and associated Tinn granite [5,68]. For most of the A-type granites the depleted mantle Nd model ages decrease slightly westwards from ca. 1.6 Ga in Subdomain C to 1.4 Ga in Subdomain B [5], which can be attributed to a progressive increase in the proportion of depleted mantle component in the source of the granitic magmas westwards [5]. In contrast, [36] suggested that the Feda mafic facies or the Valldal and Gjuve–Morgedal metabasalts could be the source of the granitic magmas, suite based on their similar Sr and Pb compositions; this model was disputed by [44]. There is, however, a general consensus that mafic underplating is a major heat source for crustal anatexis events in the region [5,18,19,44,45]. Events with input of new, mantle-derived material into the deep crust in the region have been documented at 1.13 Ga, 1.21 Ga and 1.32 Ga, and possibly also at ca. 0.97 Ga [15,20,36].

Sveconorwegian pegmatites can be divided into two age groups: Group I 1100–1030 Ma and Group II 930–890 Ma [30]. The Group I pegmatites formed during a transpressional regime while the Group II pegmatites formed at a time of extension [30]. Like the post-tectonic granite suite, the Group II pegmatites melting event is thought to be related to mafic underplating [30]. The age relations between the pegmatites studied and their neighboring granites are listed in Table 2.

**Table 2.** Age relations of pegmatites and neighboring granites. RHT Sector = Rogaland-Hardangervidda-Telemark Sector ([16,30,45,69–73]).

| Pegmatite Field | Subdomain | Pegmatite | Pegmatite Age (Ma) | Neighboring Granite | Age of Neighboring Granite (Ma) | Minimum Age Difference (Ma) | Pegmatite Group |
|---|---|---|---|---|---|---|---|
| Froland | A (Bamble Sector) | Gloserheia | 1060 +8/−6 [69] | Herefoss | 949 ± 6 [45] | 99 | 1 |
| Tørdal | B (RHT Sector) | Skardsfjell | 892.7 ± 8.8 [30] | Tørdal-Treungen | 946 ± 4 [30] | 41 | 2 |
| Tørdal | B (RHT Sector) | Upper Høydalen | 905.0 ± 2.4 [30] | Tørdal-Treungen | 946 ± 4 [30] | | 2 |
| Evje-Iveland | B (RHT Sector) | Mølland | 900.7 ± 1.8 [16] | Høvringsvatnet | 981 ± 6 [72] | 63 | 2 |
| Evje-Iveland | B (RHT Sector) | Mølland | 906 ± 9 [70] | Høvringsvatnet | 981 ± 6 [72] | 64 | 2 |
| Evje-Iveland | B (RHT Sector) | Frikstad | 910 ± 14 [71] | Høvringsvatnet | 981 ± 6 [72] | 61 | 2 |
| Evje-Iveland | B (RHT Sector) | Steli | 910.2 ± 7.1 [16] | Høvringsvatnet | 981 ± 6 [72] | 52 | 2 |
| Østfold | C (Idefjord 'Terrane') | Halvorsrød | 902.9 ± 1.7 [30] | Iddefjord | 918 ± 7 [73] | 6 | 2 |
| Østfold | C (Idefjord 'Terrane') | Karlshus | 906.3 ± 5.9 [16] | Iddefjord | 918 ± 7 [73] | ages overlap | 2 |
| Østfold | C (Idefjord 'Terrane') | Vinter-gruben | 908.9 ± 1.4 [16] | Iddefjord | 918 ± 7 [73] | ages overlap | 2 |
| Østfold | C (Idefjord 'Terrane') | Herre-bøkasa | 911.8 ± 8.7 [30] | Bohus | 922 ± 5 [74] | ages overlap | 2 |

The *Froland pegmatite field* is located in Subdomain A (Figure 1) and contains more than 105 large (>1000 m³) pegmatite bodies distributed over a NNE-SSW-trending area that measures 20 by 5 km [29] (Figure 2). The field lies approximately parallel to the Sveconorwegian shear zone that separates Subdomains A and B (Figure 2). The host rocks of the pegmatites belong to an isoclinally folded sequence of banded biotite-hornblende gneisses of volcanosedimentary origin, metamorphosed at upper amphibolite to granulite-facies metamorphism conditions at 1145 Ma to 1125 Ma [35,48,75]. The Herefoss pluton

(dated to $926 \pm 8$ Ma [18] or $949 \pm 6$ Ma [45]) and the undated Holtebu granite [65,76] occur adjacent to the south-western part of the field. The granite is significantly younger than the SE-dipping shear zone which it cross-cuts, but is itself cut by the brittle Porsgrunn–Kristiansand fault, with downthrow and ca. 2 km southwesterly displacement of the SE block [65]. No age data exist for the Froland pegmatites, but the Gloserheia pegmatite dated to $1060 +8/-6$ Ma [69] is considered to be of the same chemical and genetic type [28,29]. The mineralogy, mineral chemistry and structures of the Froland pegmatites are described in detail in e.g., [28,77–82]. In this study, K-feldspar from the Lille Kleivmyr and Sønnristjern pegmatites in the northern part of the area shown in Figure 2, and a feldspar sample from the Herefoss granite have been analyzed. Whole-rock Pb isotope data, and an analysis of K-feldspar from one granite sample by TIMS were published by [18].

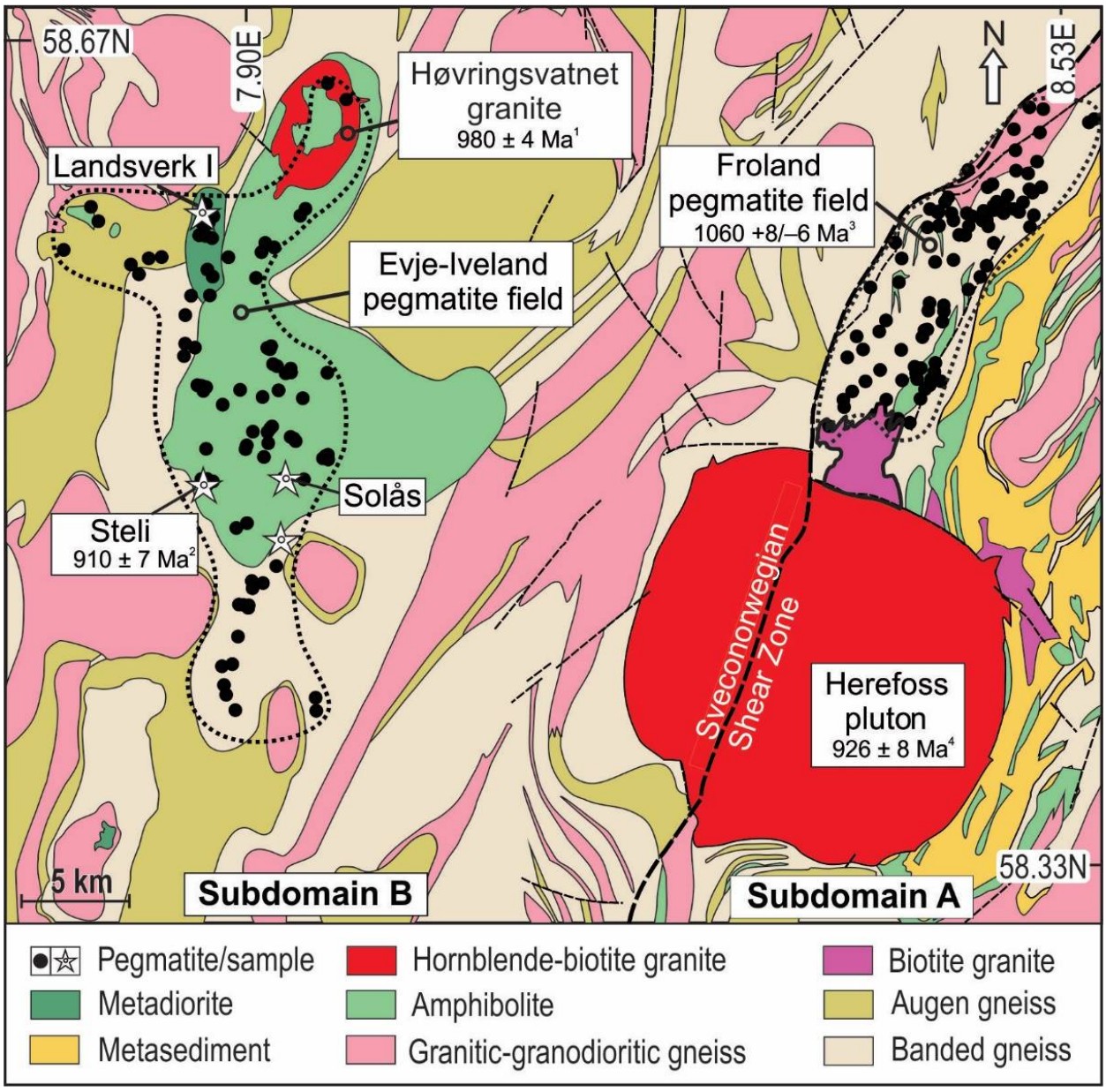

**Figure 2.** Geological map with the locations of the Froland and Evje–Iveland pegmatite fields after [29]. References of emplacement ages: [1] = [72], [2] = [16], [3] = [69], and [4] = [18]. The location of area is shown in Figure 1.

The *Evje–Iveland pegmatite field* is located in Subdomain B (Figure 1). It is approximately 30 × 10 km trending from N to S with more than 400 large (>1000 m$^3$) pegmatite bodies [29] (Figure 2). The pegmatites are hosted by banded amphibole-gneiss (1459 ± 8 Ma Vånne banded gneiss), gabbroic amphibolites of the Iveland–Gautestad mafic intrusion (1285 ± 8–1271 ± 12 Ma) and the Flåt–Mykleås metadiorite (1034 ± 2 Ma; [20]). The amphibolites are thought to be the melt source of the pegmatite melts formed by partial melting at amphibolite facies conditions [29,72]. The heat source for the melting is thought to be mafic underplating [29]. The Høvringsvatnet granite (980 ± 4 Ma [72]) is located at the northeastern margin of the pegmatite field. The pegmatites have been dated to ages around 910 Ma [16,71], see details in Table 2. The mineralogy of the pegmatites was described by [29,72,77,78,81,83–88]. In this study, K-feldspar was analyzed from the Solås, Landsverk and Steli pegmatites, and from a sample of the Høvringsvatnet granite.

The *Tørdal pegmatite field* is also situated in Subdomain B. It consists of more than 300 large (>1000 m$^3$) pegmatite bodies covering an area of 12 km × 5 km (Figure 3). The pegmatites are hosted mainly by amphibolite and metagabbro belonging to the undated Mesoproterozoic volcanic-sedimentary Nissedal supracrustal series, which is probably correlated with the ca. 1500 Ma Rjukan Group of the Telemark Supergroup. The pegmatite field is bordered to the south by the 946–952 Ma [30,45] Tørdal–Treungen granite. The pegmatite field can be divided into three zones based on the K-feldspar coloring (pink, white and green [89]). The K-feldspar colors correlate well with the degree of fractionation of the pegmatites e.g., the most evolved pegmatites occur in the green zone. Increasing fractionation is not correlated with distance from the intrusive contact of the granites. Two unpublished pegmatite ages exist: the Upper Høydalen pegmatite at 905 ± 2 Ma and the Skardsfjell pegmatite at 893 ± 9 Ma (both in [30]). The mineralogy of the pegmatites has been described by [90–92]. In this study, K-feldspar has been analyzed from three pegmatites (Kleppe quarry, Svåheii 3, Upper Høydalen), from a pegmatitic body of the Tørdal-Treungen granite, and from the granite itself.

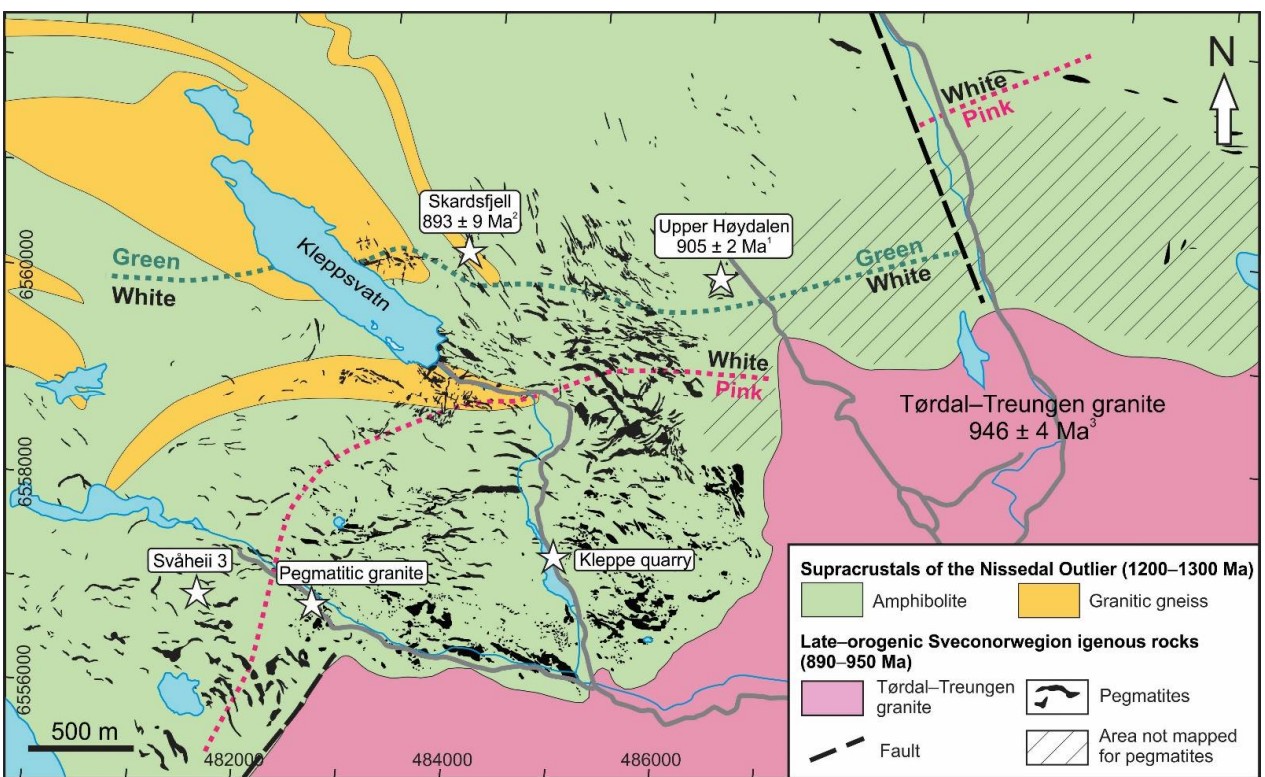

**Figure 3.** Geological map with the locations of the Tørdal pegmatite field. References of emplacement ages: [1], [2], [3] = [30]. Location of map is shown in Figure 1.

The *Flesberg pegmatite field* is situated in Subdomain B, near the boundary to Subdomain C. It is approximately 65 km long in NE-SW direction, and 10 km wide. It is hosted mainly by granitic to granodioritic gneisses but pegmatites are also found in metasandstone and biotite gneiss. No potential parental granite for the pegmatite field is exposed in the area, the closest granite being the Flå granite (928 ± 3 Ma [35]), more than 20 km away from the pegmatite field. The mineralogy of the pegmatites is described in [30]. K-feldspar has been analyzed from the Bjertnes pegmatite (Figure 4). This pegmatite is located close to the thrust zone called the Kongsberg–Telemark Boundary Zone [49]. K-feldspar Pb isotope analyses by TIMS from the Flå granite were published by [5].

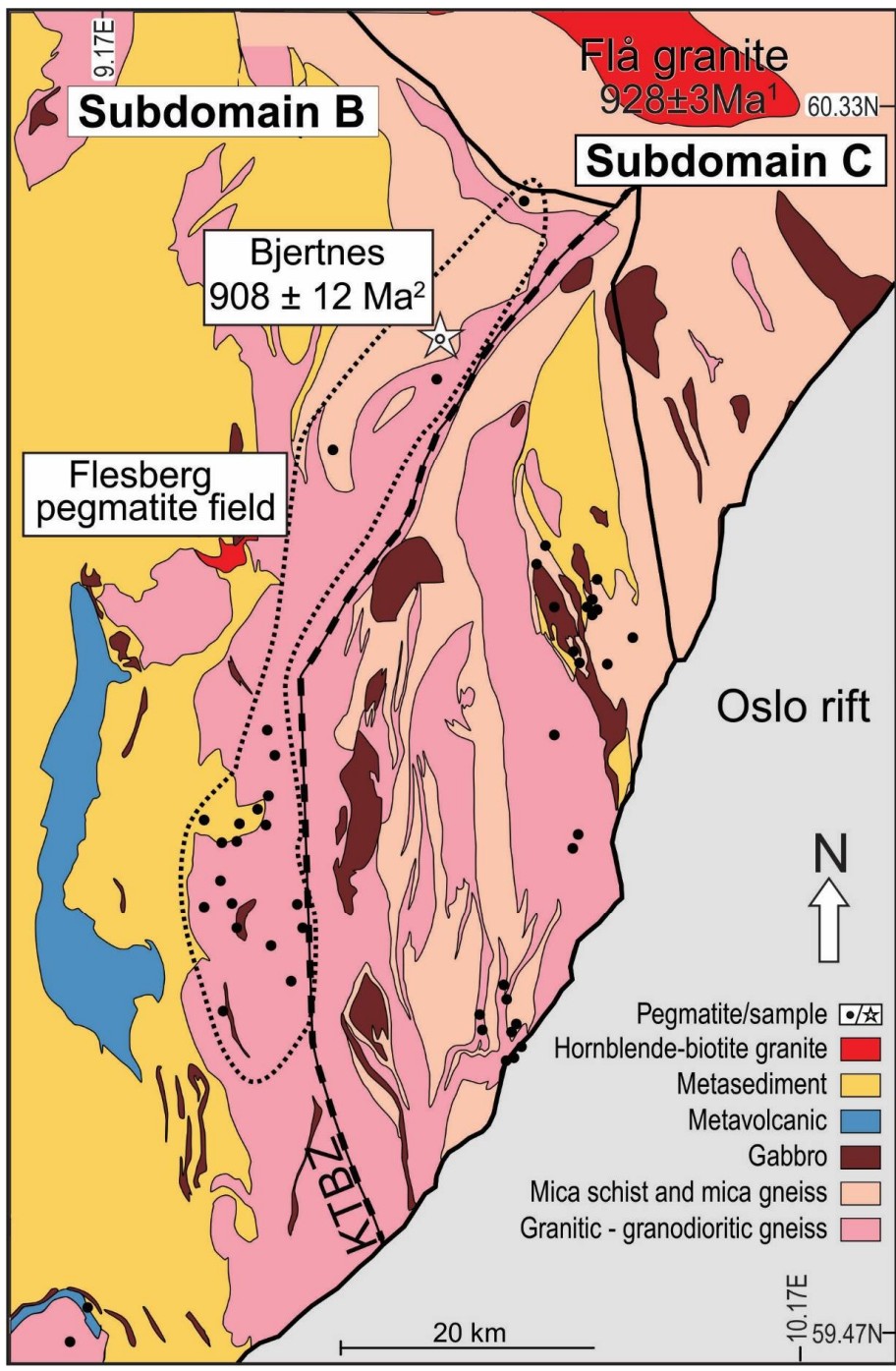

**Figure 4.** Geological map with the location of the Flesberg pegmatite field. KTBZ = Kongsberg-Telemark Boundary Zone [49]. References of emplacement ages: [1] = [35] and [2] = [30]. Modified after NGU 1:250,000 Geological map of Norway. Location of area is shown in Figure 1.

The *Østfold pegmatite field* is located in Subdomain C, where it is hosted by Mesoproterozoic gneisses of the Østfold Gneiss Complex [93]. The pegmatite field stretches approximately 80 km between Oslo and Halden containing around 150 large (>1000 m$^3$) pegmatite bodies (Figure 5). The Østfold Gneiss Complex and its continuation in southwestern Sweden was intruded by the late Sveconorwegian A-type Østfold (Iddefjord)—Bohus batholith, which has been dated to 918 ± 7 Ma by Rb-Sr [73] and 920 ± 7 Ma by TIMS-ID U-Pb [74], some pegmatites occur within the batholith. Several pegmatites are dated from the Østfold pegmatite field: 902.9 ± 1.7 Ma Halvorsrød [30], 906 ± 6 Ma Karlshus [16], 908.9 ± 1.4 Ma Vintergruben [16], 922 ± 3 Grebbestad [16] and 912 ± 9 Ma Herrebøkasa [30]. The mineralogy of the Østfold pegmatites was described in [94], who found a geographical distribution of pegmatite types, with rare mineral rich pegmatites restricted to the area close to the intrusive contact of the Østfold–Båhus granite. This distribution pattern was later confirmed from a larger set of observations by [95]. K-feldspar from the Herrebøkasa pegmatite was analyzed in the present study. This pegmatite occurs close to the border of the Bohus–Iddefjord batholith.

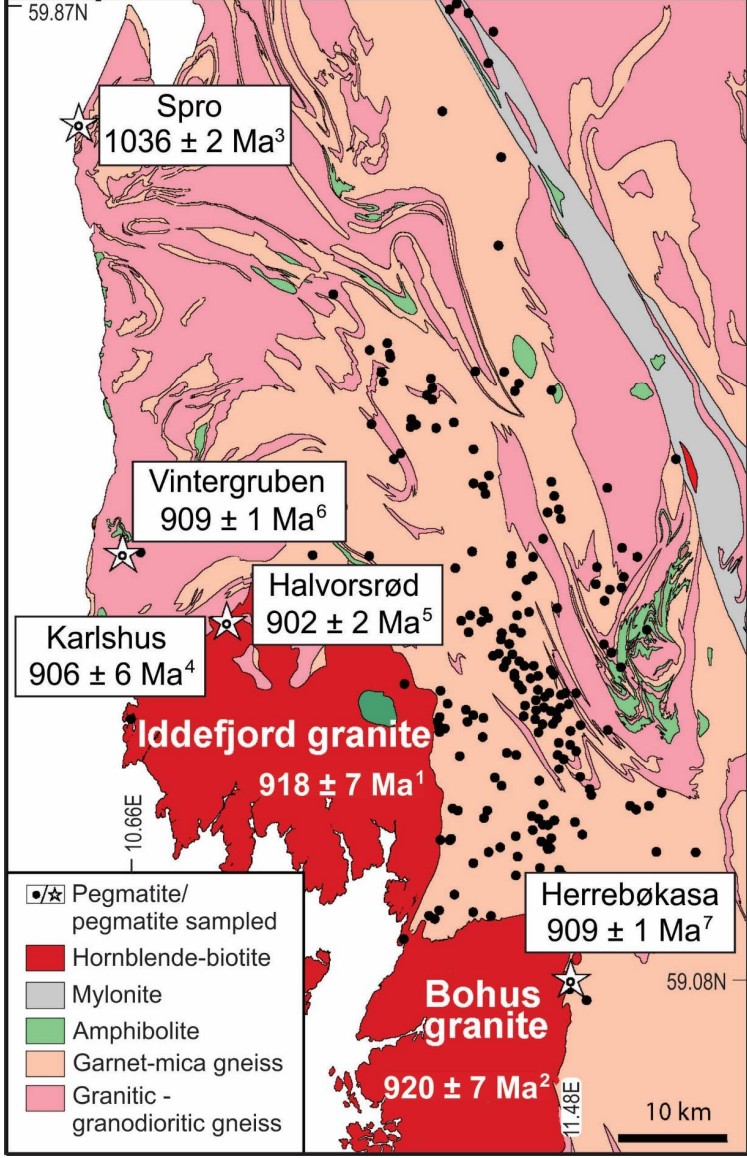

**Figure 5.** Geological map with the locations of the Østfold pegmatite field in Subdomain C (Figure 1). References of emplacement ages: [1] = [73], [2] = [74], [3], [5], [7] = [30], and [4], [6] = [16]. Modified after NGU 1:250,000 Geological map of Norway. Location of map is shown in Figure 1.

### 3. Analytical Methods

Pb isotope analysis of K-feldspar was done by laser ablation inductively coupled plasma mass spectrometry (LA-ICP-MS), using a Nu Plasma HR multi-collector mass spectrometer equipped with a Cetac LSX-213G2+ laser microprobe at the Department of Geosciences, University of Oslo. The ablation conditions were: beam diameter 200 μm (aperture imaging mode) and pulse frequency 10 Hz. Masses 208–201 were measured simultaneously in Faraday cups using the Nu Plasma U-Pb collector block. These masses correspond to Pb: 208, 207, 206; Pb and Hg: 204; Tl: 205, 203; Hg: 202, 201. Each ablation was preceded by 30 s of background measurement, which was subsequently subtracted from the signal. Any contribution from $^{204}$Hg on the total $^{204}$(Pb + Hg) signal was corrected for using the natural $^{204}$Hg/$^{202}$Hg = 0.23007 [96] and the measured 202 mass signal. To correct for mass discrimination in the mass spectrometer, a thallium solution with known isotopic composition was aspired through a desolvating nebulizer into the transport line between ablation cell and plasma torch and raw lead isotope ratios were normalized to a constant $^{205}$Tl/$^{203}$Tl ratio (2.3889) using an exponential law, e.g., [97]. Data reduction was done in the Nu Plasma calculation software.

### 4. Results

Individual LA-ICPMS analyses of K-feldspar scatter by more than the analytical error of the individual point, most likely due to small-scale heterogeneity, exsolution of plagioclase lamellae, and sub-surface inclusions with higher U content (and hence higher U/Pb ratio and more radiogenic present-day Pb). The results are summarized in Table 3 and the complete dataset is provided in the Supplementary Material (Table S1).

#### 4.1. Froland

The two Froland pegmatites analyzed (Figure 6a,b) have $^{206}$Pb/$^{204}$Pb of 17.380 and 17.322, but $^{207}$Pb/$^{204}$Pb overlapping within error at 15.516 ± 0.016. These are significantly more radiogenic compositions than that of feldspar in the Herefoss granite at $^{206}$Pb/$^{204}$Pb = 17.065, $^{207}$Pb/$^{204}$Pb = 15.468 (present study) and $^{206}$Pb/$^{204}$Pb = 16.697, $^{207}$Pb/$^{204}$Pb = 15.445 [18]. The pegmatite feldspar samples are not co-linear with whole-rock samples of the Herefoss granite [18] along a 926 Ma reference line corresponding to the age of the granite. There is, however, marginal overlap within error with a 926 Ma reference line through the whole-rock samples of the more radiogenic Holtebu granite [18], which has relatively higher $^{207}$Pb/$^{204}$Pb. The pegmatite K-feldspar analyses show marginal overlap with K-feldspar from the 1120 Ma Ubergsmoen charnockite intrusion, whose source is thought to represent the Palaeoproterozoic crust in Subdomain A [17]. The feldspars from the granite and the Froland pegmatites have $^{208}$Pb/$^{204}$Pb below 36.7, overlapping with feldspar in the Ubersgmoen charnockite (Figure 6c, data from [17]). The whole-rock samples of the Herefoss granite have very variable Th/U ratios, and hence variation in present-day $^{208}$Pb/$^{204}$Pb that is not correlated with $^{206}$Pb/$^{204}$Pb [18].

**Table 3.** Lead isotope data of K-feldspars from Sveconorwegian pegmatites and granites.

| Area | Sample | Locality | Type | $^{206}$Pb/$^{204}$Pb | 2 σ | $^{207}$Pb/$^{204}$Pb | 2 σ | $^{208}$Pb/$^{204}$Pb | 2 σ | *n* | Method | Reference |
|---|---|---|---|---|---|---|---|---|---|---|---|---|
| Froland-Herefoss | | | | | | | | | | | | |
| | 2109402 | Lille Kleivmyr | K-fsp | 17.379 | 0.011 | 15.520 | 0.010 | 36.676 | 0.023 | 9 | LA-ICPMS | This work |
| | 2009407 | Sønnristjern | K-fsp | 17.322 | 0.011 | 15.511 | 0.010 | 36.599 | 0.023 | 10 | LA-ICPMS | This work |
| | 26051712 | Herrefoss granite | K-fsp | 17.065 | 0.020 | 15.468 | 0.018 | 36.562 | 0.042 | 3 | LA-ICPMS | This work |
| | 107KF | Herefoss granite | K-fsp | 16.697 | 0.017 | 15.445 | 0.015 | 36.478 | 0.036 | | TIMS | [26] |
| | 107wr | Herefoss granite | WR | 17.626 | 0.018 | 15.502 | 0.016 | 37.654 | 0.100 | | TIMS | [26] |
| Evje-Iveland | | | | | | | | | | | | |
| | 59295 | Solås | K-fsp | 17.390 | 0.024 | 15.508 | 0.010 | 36.697 | 0.023 | 10 | LA-ICPMS | This work |
| | 09070819 | Solås | K-fsp | 17.343 | 0.012 | 15.520 | 0.010 | 36.740 | 0.023 | 9 | LA-ICPMS | This work |
| | 12070802 | Landsverk | K-fsp | 17.137 | 0.011 | 15.489 | 0.010 | 36.509 | 0.023 | 10 | LA-ICPMS | This work |
| | 07070802 | Steli | K-fsp | 17.555 | 0.012 | 15.534 | 0.010 | 36.810 | 0.023 | 9 | LA-ICPMS | This work |
| | 22051701 | Høvringsvatnet granite | K-fsp | 16.637 | 0.056 | 15.446 | 0.045 | 36.317 | 0.130 | 5 | LA-ICPMS | This work |
| | | Høvringsvatnet granite | WR | 17.142 | 0.013 | 15.471 | 0.019 | 37.459 | 0.059 | | TIMS | [5] |
| Tørdal | | | | | | | | | | | | |
| | 07061607 | Kleppe quarry | K-fsp | 17.266 | 0.011 | 15.503 | 0.010 | 36.703 | 0.023 | 10 | LA-ICPMS | This work |
| | 05061610 | Svåheii 3 | K-fsp | 17.307 | 0.011 | 15.506 | 0.010 | 36.753 | 0.023 | 10 | LA-ICPMS | This work |
| | 23091503 | Upper Høydalen | K-fsp | 17.331 | 0.011 | 15.504 | 0.010 | 36.736 | 0.023 | 10 | LA-ICPMS | This work |
| | 23091502 | Upper Høydalen | K-fsp | 17.326 | 0.014 | 15.503 | 0.013 | 36.728 | 0.030 | 6 | LA-ICPMS | This work |
| | 05061621 | Pegmatitic granite | K-fsp | 17.252 | 0.011 | 15.506 | 0.010 | 36.723 | 0.023 | 10 | LA-ICPMS | This work |
| | 20091501 | Tørdal granite (Skjeggefoss quarry) | K-fsp | 17.338 | 0.011 | 15.512 | 0.010 | 36.832 | 0.023 | 10 | LA-ICPMS | This work |
| | | Tørdal WR | WR | 19.503 | 0.046 | 15.593 | 0.020 | 43.181 | 0.072 | | TIMS | [5] |
| | | Treungen WR | WR | 17.386 | 0.014 | 15.474 | 0.020 | 41.409 | 0.059 | | TIMS | [5] |
| Flesberg | | | | | | | | | | | | |
| | Bjertnes | KFS | K-fsp | 17.318 | 0.011 | 15.515 | 0.010 | 36.941 | 0.023 | 10 | LA-ICPMS | This work |
| Østfold | | | | | | | | | | | | |
| | 14101602 | Herrebøkasa | K-fsp | 16.908 | 0.011 | 15.486 | 0.010 | 36.755 | 0.023 | 10 | LA-ICPMS | This work |
| | 14101610 | Iddefjord granite | K-fsp | 16.953 | 0.012 | 15.492 | 0.010 | 36.737 | 0.024 | 5 | LA-ICPMS | This work |

K-feldspar data are given as error-weighted means [98] with 2 s error of the mean. WR: Published whole-rock data by TIMS. Additional data on K-feldspar separates from the Iddefjord and Flå granites by TIMS on mineral separates were published by [5].

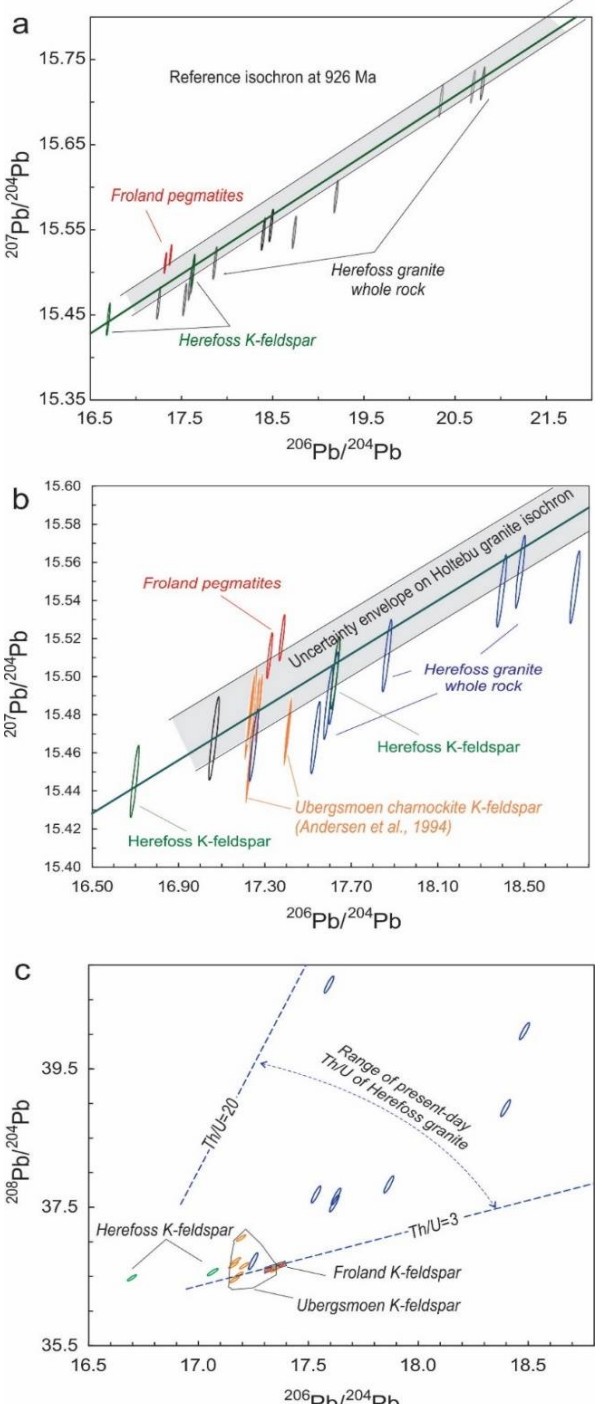

**Figure 6.** Lead isotope data of the Froland pegmatites (this work) and the Herefoss granite (this work and [18]). Analyses are represented by 2 σ error ellipses, assuming an error correlation of 0.9 (which also applies to Figures 7–10). (**a**) Granite whole rock and feldspar samples shown relative to a 926 Ma reference mineral isochron from [18], and a shaded field representing the expected range of variation of Pb in the Holtebu granite at 926 Ma. (**b**) Detail of *a*, also showing feldspar Pb analyses from the Ubersgmoen charnockite [17]. (**c**) $^{208}$Pb/$^{204}$Pb vs. $^{206}$Pb/$^{204}$Pb for Froland pegmatites and Herefoss granite. The whole-rock samples of the Herefoss granite show a wide variation in their U/Th ratio, resulting in a scatter of present-day compositions that can be explained by radiogenic accumulation from a common initial at atomic Th/U ratios between 3 and 20.

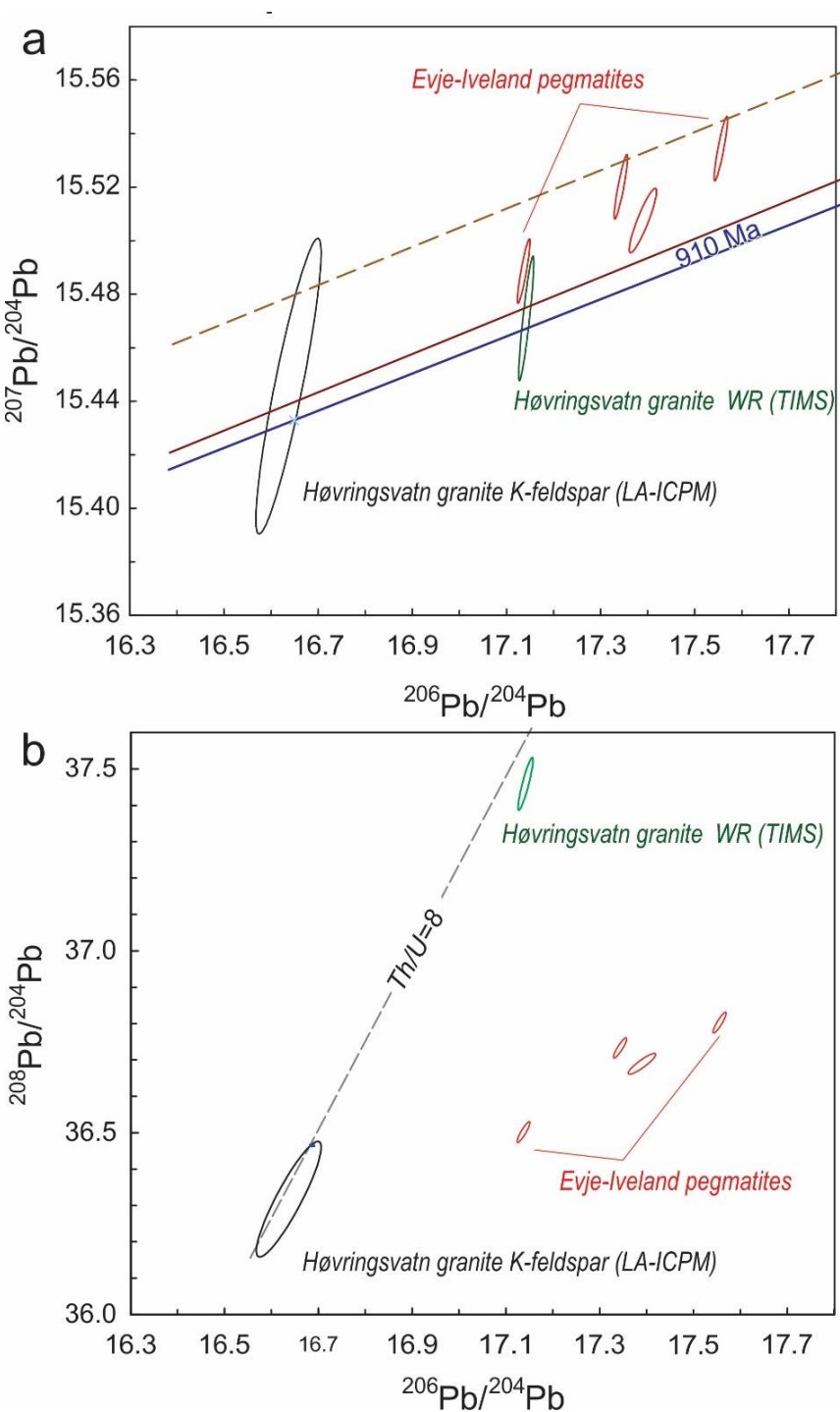

**Figure 7.** Evje–Iveland pegmatites and Høvringsvatn granite feldspar. (**a**) $^{207}$Pb/$^{204}$Pb vs. $^{206}$Pb/$^{204}$Pb of feldspars in pegmatites and Høvringsvatn granite (granite data from this study and [5]). K-feldspars from the pegmatites are on moderately more radiogenic than the granite but will overlap with the range of variation of the Høvringsvatn granite at 910 Ma. (**b**) $^{208}$Pb/$^{204}$Pb vs. $^{206}$Pb/$^{204}$Pb. The whole rock Pb isotope composition of the Høvringsvatn granite analyzed by [5] could have developed from the feldspar of the granite analyzed in this study, but at a very high Th/U ratio. Pegmatite feldspars fall off this trend towards high $^{206}$Pb/$^{204}$Pb/low $^{208}$Pb/$^{204}$Pb.

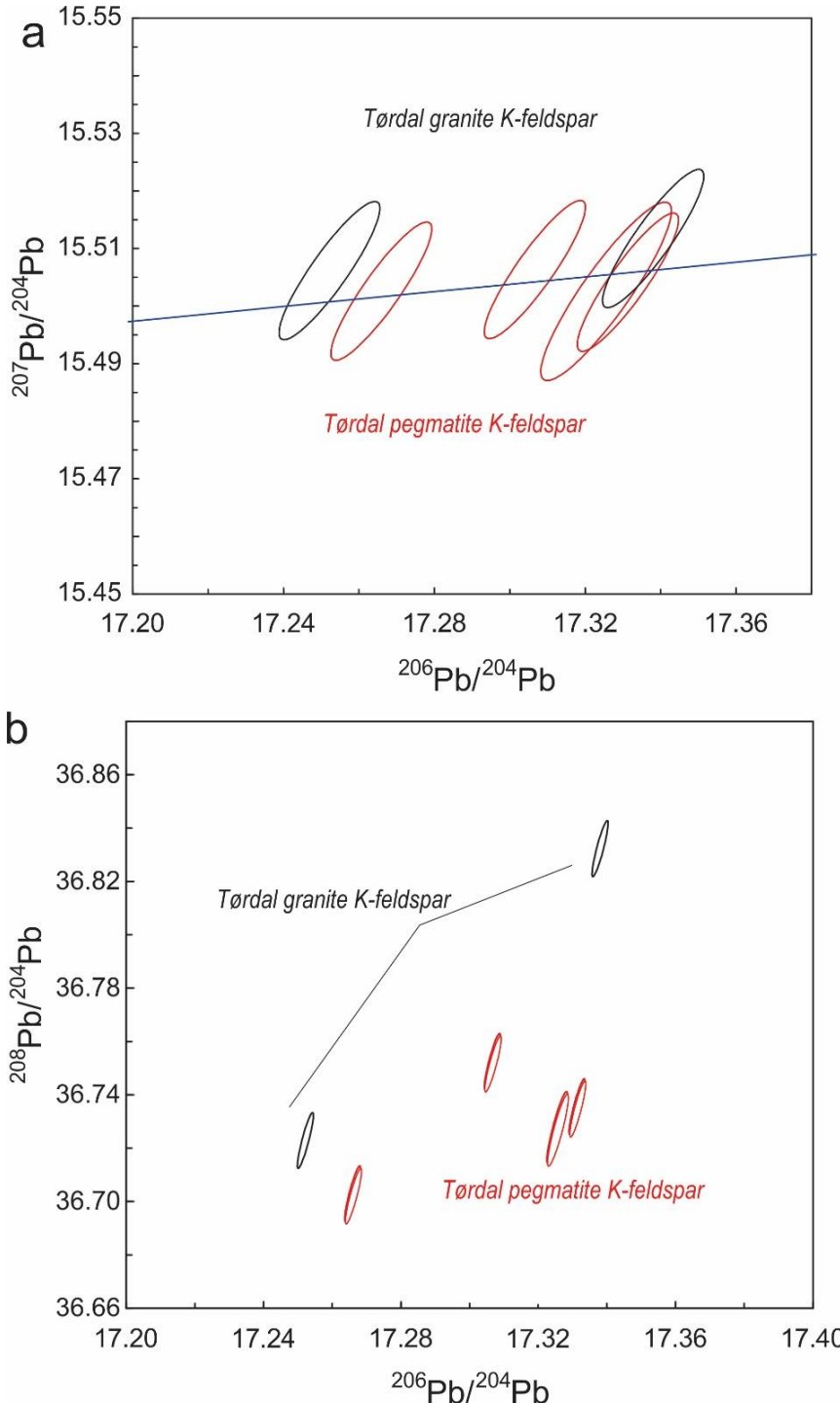

**Figure 8.** Lead isotopes in Tørdal pegmatite and granite feldspar. (**a**) $^{207}Pb/^{204}Pb$ vs. $^{206}Pb/^{204}Pb$ of feldspars in pegmatites and Tørdal–Treungen granite, relative to a 920 reference line. Whole-rock samples of Tørdal and Treungen granite subfacies plot off the scale of this diagram and are not colinear with the feldspar samples along a present-day isochron with reasonable age [5]. (**b**) $^{208}Pb/^{204}Pb$ vs. $^{206}Pb/^{204}Pb$ of Tørdal K-feldspars. One of the granite feldspars overlaps with the pegmatites in $^{208}Pb/^{207}Pb$, the other not.

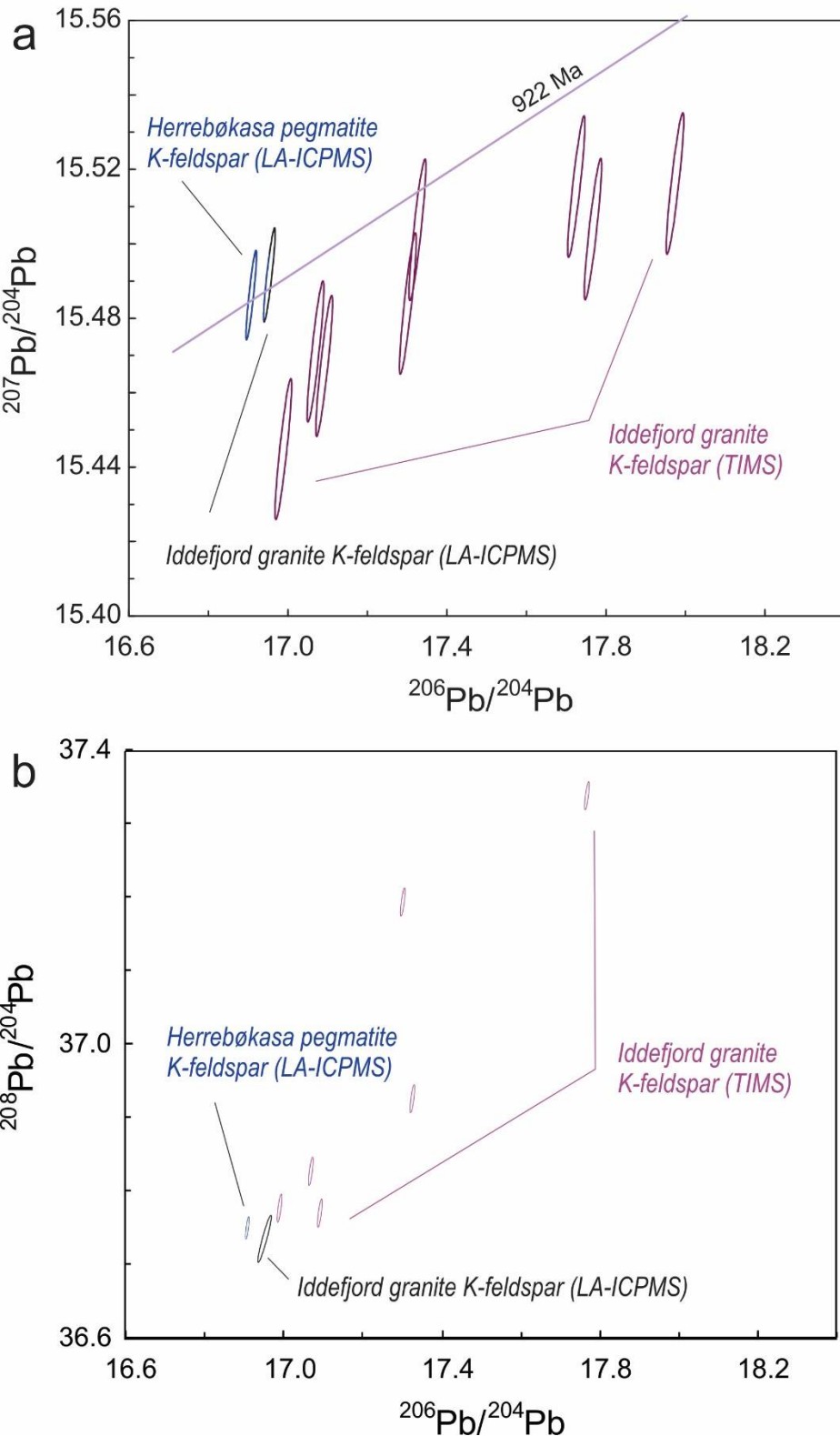

**Figure 9.** Lead isotopes in the Herrebøkasa pegmatite and Iddefjord granite feldspar. (**a**) $^{207}$Pb/$^{204}$Pb vs. $^{206}$Pb/$^{204}$Pb of feldspar from the pegmatite compared to feldspar in the Iddefjord granite (LA-ICPMS data from this study and TIMS analyses from [5]), compared to a 922 Ma reference line through the granite feldspar. (**b**) $^{208}$Pb/$^{204}$Pb vs. $^{206}$Pb/$^{204}$Pb, showing the large spread in $^{208}$Pb/$^{204}$Pb in the granite feldspars, suggesting high Th/U ratio in the source region.

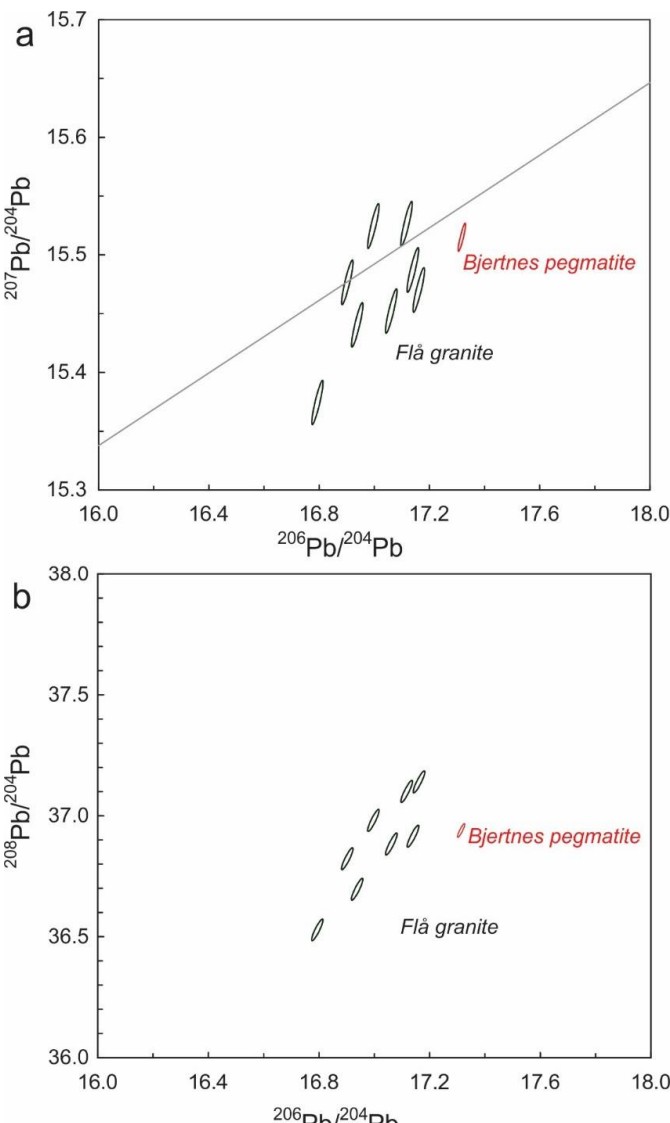

**Figure 10.** K-feldspar data from the Bjertnes pegmatite (this work) and K-feldspar separates by TIMS for the Flå granite [5]). The broken line in a is a 920 Ma reference line. The Bjertnes pegmatite K-feldspar is slightly more radiogenic in $^{206}$Pb/$^{204}$Pb but overlap with the granite feldspars in $^{207}$Pb/$^{204}$Pb (**a**) and $^{208}$Pb/$^{204}$Pb (**b**).

### 4.2. Evje-Iveland

The Evje–Iveland pegmatites range from $^{206}$Pb/$^{204}$Pb = 17.137 to 17.555, with $^{207}$Pb/$^{204}$Pb between 15.489 and 15.537 (Figure 7a). The samples are colinear with each other along a reference line with an age of 910 Ma, and within error also with the less radiogenic feldspar sample from the granite, and the granite whole-rock analysis reported by [5]. The pegmatite feldspars show variation in excess of analytical error in $^{208}$Pb/$^{204}$Pb (Figure 7b). The present-day composition of the Høvringsvatn granite [5] can have developed from the feldspar Pb at a high Th/U ratio, shown for a 910 Ma reference isochron with Th/U = 8 in Figure 7b.

### 4.3. Tørdal

K-feldspar from the Tørdal pegmatites show full overlap with feldspar from the Tørdal–Treungen granite, forming an array with $^{206}$Pb/$^{204}$Pb = 17.252 to 17.338, at $^{207}$Pb/$^{204}$Pb overlapping within error at 15.506 (Figure 8a). This is, however, significantly different from the published data on whole-rock samples of the Tørdal granite subfacies at

$^{206}Pb/^{204}Pb = 19.505$, $^{207}Pb/^{204}Pb = 15.593$ and the Treungen granite subfacies at $^{206}Pb/^{204}Pb = 17.386$, $^{207}Pb/^{204}Pb = 15.474$ [5], neither of which are colinear within error with the feldspar analyses from the Tørdal–Treungen granite and Tørdal pegmatites along 900 Ma to 920 Ma reference lines. $^{208}Pb/^{204}Pb$ in the Tørdal pegmatites varies between 36.70 and 36.50. The feldspar of the pegmatitic granite falls in the same range, the other granite sample has a more orogenic composition at 36.83 (Figure 8b).

### 4.4. Østfold

Feldspar samples from the Herrebøkasa pegmatite and a sample of Østfold (Iddefjord) granite give nearly overlapping Pb isotope compositions at $^{207}Pb/^{204}Pb = 16.908$, $^{207}Pb/^{204}Pb = 15.486$, $^{208}Pb/^{204}Pb = 36.755$ and $^{207}Pb/^{204}Pb = 16.953$, $^{207}Pb/^{204}Pb = 15.492$, $^{208}Pb/^{204}Pb = 36.737$, respectively (Figure 9a). In comparison, TIMS Pb isotope data from feldspar separates of the Østfold granite [5] shows relatively large scatter in $^{206}Pb/^{204}Pb$ and $^{207}Pb/^{204}Pb$ towards more radiogenic values, some of which may be due to inclusions in the K-feldspar, but still show marginal overlap with the new data along a 922 Ma reference line. Feldspar in both granite and pegmatite have higher $^{208}Pb/^{204}Pb$ than the other examples in this study but overlaps with the unradiogenic end of the range of variation of feldspar separates from the granite (Figure 9b).

### 4.5. Flå Granite and Bjertnes Granite Pegmatite

TIMS Pb- isotope analyses of K-feldspars of the Flå granite show scatter in excess of analytical uncertainty, with $^{206}Pb/^{204}Pb$ in the range 16.8 to 17.2, $^{207}Pb/^{204}Pb$ from 15.37 to 15.53 and $^{208}Pb/^{204}Pb$ between 36.5 and 37.1 (Figure 10a,b, data from [5]). Whereas this scatter may be real, a possible effect of inclusions in the grains dissolved for TIMS analysis cannot be disregarded in these feldspars. The LA-ICPMS analysis of K-feldspar from the Bjertnes pegmatite has $^{206}Pb/^{204}Pb = 17.318$, which is marginally more radiogenic than the granite feldspars, but $^{207}Pb/^{204}Pb$ and $^{208}Pb/^{204}Pb$ fall within the range of the granite feldspar separates.

## 5. Discussion

### 5.1. Brief Background for Interpretation of the Feldspar and Whole-Rock Pb Isotope Data

In Fennoscandia, a significant volume of juvenile crust formed at in the period 2.10 Ga to 1.86 Ga [99]. This protolith was modified in several events of intra-crustal magmatism in late Palaeoproterozoic and Mesoproterozoic time [33,44,46,99], in some of which mantle-derived mafic melts were emplaced in the deep to middle crust [15,20,100]. For the present purpose, the average, initial Pb isotope composition of the juvenile crustal protolith can be approximated by the Stacey and Krames second-stage model [101] at 1.9 Ga (i.e $^{206}Pb/^{204}Pb = 15.363$, $^{207}Pb/^{204}Pb = 15.239$, $^{208}Pb/^{204}Pb = 34.999$). Subsequent internal crustal differentiation processes and introduction of mantle-derived material will cause heterogeneities in U/Pb and Th/U ratios, and thereby diverging trends of Pb isotope composition with time.

Figure 11a shows the schematic evolution of crustal domains with low and high, time-integrated U/Pb ratios between formation of the protolith at $t_1$ (e.g., 1900 Ma) and the time of formation of late Sveconorwegian granitic magmas at $t_2$ (i.e., 900–1000 Ma). At $t_2$, domains of age $t_1$ with different U/Pb ratios (or more correctly: $^{238}U/^{204}Pb$ ratios) will be colinear with each other and the common initial Pb composition at $t_1$ along a straight line which is technically known as a palaeoisochron. Partial melting of domains with variable U/Pb since $t_1$ will generate magmas whose Pb isotopic compositions at $t_2$ fall on the palaeoisochron, and each of these will in turn be the initial ratio of an isochron with age $t_2$, defined by systems sharing initial Pb isotopic composition at $t_2$, but with variable U/Pb since $t_2$. Granites or pegmatites with a range of initial compositions along the palaeoisochron at $t_2$ will then give rise to a band of parallel isochrons with age $t_2$. K-feldspar in these granites will not contain U or Th, and remain stationary at the initial ratio. The expected result is a band of feldspar Pb plotting along the relevant palaeoisochron

trend in the diagram. Some complications are caused by introduction of less radiogenic Pb from mantle-derived material at $t_2$ (or at any other time between $t_1$ and $t_2$). This will shift the composition of affected systems as indicated by arrows in Figure 11a. The result will be increased spread of the present-day K-feldspar data away from the palaeoisochron.

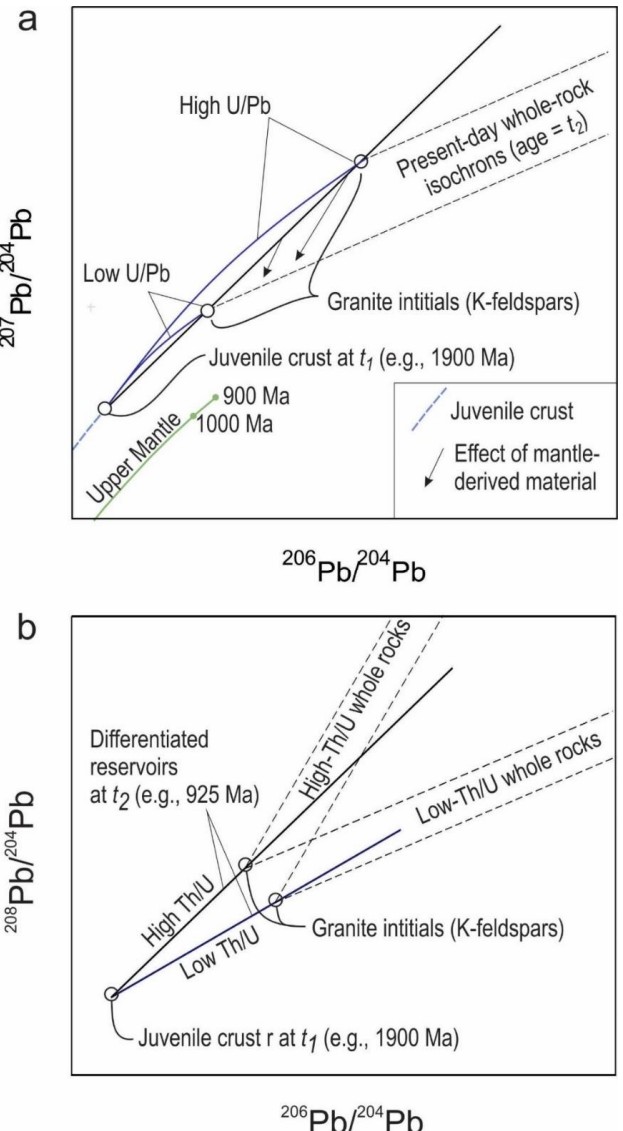

**Figure 11.** Sketches of two- and three-stage Pb isotope evolution of relevance for interpretation of the granite and pegmatite K-feldspar Pb. (**a**) $^{207}Pb/^{204}Pb$ vs. $^{206}Pb/^{204}Pb$. The initial Pb isotope composition of the regional crustal protolith at the age of its formation ($t_1$, assumed to be 1.9 Ga) is assumed to correspond to a global model for juvenile crust given by the Stacey and Kramers second stage model (blue, broken curve [101]). After formation, the crust differentiated into domains with high and low U/Pb ratios (really $^{238}U/^{204}Pb$ ratios), which evolved as closed systems until extraction of granitic melts at $t_2$. Granite whole rock system will plot on isochron lines with slope corresponding to $t_2$, and with initial ratios at the palaeoisochron of $t_1$ systems at $t_2$, which is preserved in the K-feldspars. Addition of unradiogenc material from the upper mantle, represented by the green curve will pull compositions away from the palaeoisochron, and cause scatter in K-feldspar data below this line. (**b**) $^{208}Pb/^{204}Pb$ vs. $^{206}Pb/^{204}Pb$. Crustal domains of age $t_1$ with different Th/U ratios will at $t_2$ have developed to points along lines whose slope reflects the Th/U ratio. These will in turn produce granitic melts at $t_2$, whose initial ratios preserved by K-feldspar reflect both age of the protolith and the time-integrated U/Th ratio. The slope of final stage whole-rock isochrons at present depends both on age ($t_2$) and U/Th ratio of the whole-rock.

The evolution of $^{208}$Pb/$^{204}$Pb with time depends on both the $^{238}$U/$^{204}$Pb and the $^{232}$Th/$^{238}$U ratio of the system, the latter of which is closely approximated by the atomic Th/U ratio (Figure 11b). Because of variations in the Th/U ratio, systems derived from a common initial at $t_1$ will not fall on a single palaeoisochron at in the $^{208}$Pb/$^{204}$Pb vs. $^{206}$Pb/$^{204}$Pb diagram at $t_2$, but will plot on lines whose slopes are given by:

$$\frac{\frac{^{232}\text{Th}}{^{204}\text{Pb}}(e^{\lambda_2 t_1} - e^{\lambda_2 t_2})}{\frac{^{238}\text{U}}{^{204}\text{Pb}}(e^{\lambda_8 t_1} - e^{\lambda_8 t_2})} \approx \left(\frac{\text{Th}}{\text{U}}\right)_{\text{Atomic}} \frac{(e^{\lambda_2 t_1} - e^{\lambda_2 t_2})}{(e^{\lambda_8 t_1} - e^{\lambda_8 t_2})}$$

where $\lambda_8$ is the decay constant of $^{238}$U and $\lambda_2$ that of $^{232}$Th. This composition is preserved by K-feldspar formed at $t_2$, and by the initial ratios of present-day whole-rock isochrons, whose slopes are also dependent on the Th/U ratio (Figure 11b).

### 5.2. U-Th-Pb Systematics of the Crustal Precursor(s)

The present-day, whole-rock Pb isotope compositions of Sveconorwegian, posttectonic granites across southern Norway [5] spread out along a band in the $^{207}$Pb/$^{204}$Pb vs. $^{206}$Pb/$^{204}$Pb diagram limited by lines whose slope corresponds to late Sveconorwegian ages, and whose unradiogenic end overlaps with the range of feldspars in the 1120 Ma Ubergsmoen charnockite (interpreted to be of a crustal origin) and the data on K-feldspar from the present study (Figure 12a). The K-feldspar data scatter along and below a 1900 Ma palaeoisochron at 925 Ma, suggesting that the melts from which the feldspars crystallized have been influenced by a mantle derived component. These feldspar compositions overlap completely with the range of initial compositions of the granites. On the other hand, partial melting of mantle-derived mafic rocks without significant involvement of felsic crustal material cannot account for the observed Pb isotope compositions in either pegmatites or granites. Mantle-derived mafic rocks would have Pb isotope compositions at or close to the upper mantle curve in Figure 12a (values from [102]), which is at a distinctly lower $^{207}$Pb/$^{204}$Pb level than in any of the granite or pegmatite samples.

Granite whole-rock data show a spread in $^{208}$Pb/$^{204}$Pb that is uncorrelated with $^{206}$Pb/$^{204}$Pb, indicating highly variable Th/U in the whole-rock system (Figure 6c). Similar spread is seen also in the granite and pegmatite feldspar data (Figures 6–10 and 12b), showing that the sources of melt must also have had variations in the Th/U ratio. Two trends are apparent in the summary plot of feldspar data in Figure 12b: One, comprising the feldspar data from the Froland, Evje–Iveland and Tørdal pegmatites and the Herefoss, Høvringsvatn and Tørdal–Treungen granites define an array with low slope, suggesting that the crustal precursors evolved Th/U < 3. In contrast, the Herrebøkasa and Bjertnes pegmatite feldspars, and feldspars from the Østfold and Flå granites scatter along a reference palaeoisochron with Th/U = 4, indicating that the crustal precursor in the eastern part of the province had a higher time-integrated Th/U ratio, reflecting lower U concentration and/or a higher Th concentration.

### 5.3. Granites vs. Pegmatites

Fennoscandia has a long history of internal crustal differentiation by intra-crustal anatectic magmatism, e.g., [6] and references therein. For example, the large, mainly granitic late Paleoproterozoic Transscandinavian Igneous Belt [33] formed by anatectic melting of juvenile, orogenic Paleoproterozoic crust [46]. The late Mesoproterozoic A-type granite plutons across southern Norway contain a significant proportion of material with a crustal prehistory back to the Paleoproterozoic [5,17,44] although variable amounts of early Mesoproterozoic juvenile crustal components and mid- to late Mesoproterozoic mantle-derived material have also been involved [18–20,66].

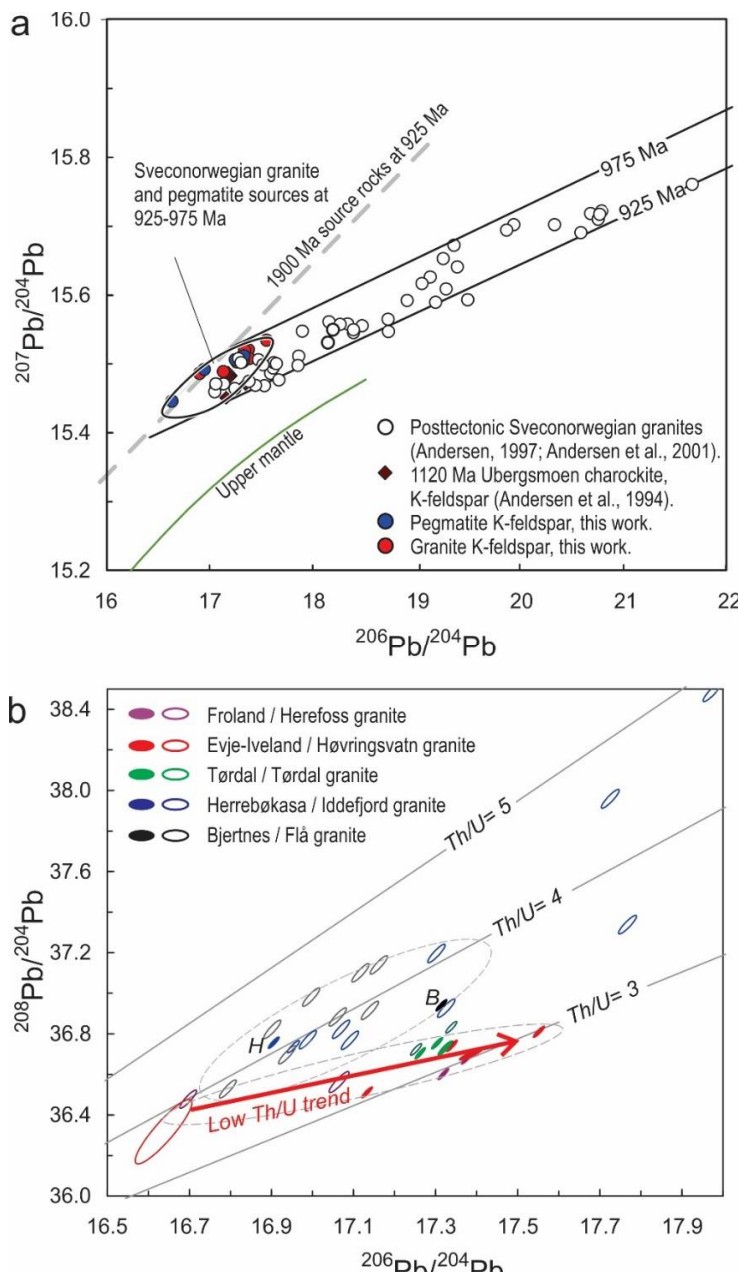

**Figure 12.** Summary plots of initial Pb isotope data of Sveconorwegian A-type granites and granite pegmatites. (**a**) $^{207}Pb/^{204}Pb$ vs. $^{206}Pb/^{204}Pb$. Present-day whole-rock Pb isotope data on the posttectonic Sveconorwegian A-type granites [5] form a band in the diagram, which overlaps the range of variation of K-feldspars in the present granite and pegmatite samples and in the 1120 Ma Ubersgmoen charnockite, thought to represent the crustal endmember of Subdomain A. A palaeoisochron at 925 Ma for 1900 Ma sources with an initial composition of the Stacey and Kramers model reservoir [101] at 1900 Ma. Feldspar Pb isotope compositions spread along and below the palaeoisochron, suggesting that Pb in both pegmatites and granites has been derived from heterogeneous crust, combined with variable input of mantle-derived material. (**b**) $^{208}Pb/^{204}Pb$ vs. $^{206}Pb/^{204}Pb$. Lead isotope compositions of granites (open) and pegmatite (filled) feldspars are shown relative to reference palaeoisochrons with $t_1 = 1900$ Ma, $t_2 = 925$ Ma and atomic Th/U ratio as indicated. Two trends may be discerned in the feldspar data: One with a low (<3) Th/U ratio, represented by the Froland, Evje–Iveland and Tørdal pegmatites and granites, and one at a higher Th/U ratio in the Herrebøkasa—Iddefjord granite and Bjertnes—Flå granite systems. H: K-feldspar from the Herrebøkasa pegmatite, B: K-feldspar from the Bjertnes pegmatite. Sources of data: The present study and [5].

When compared to previously published TIMS Pb isotope data on granites and K-feldspar separates from granites [5], the LA-ICPMS K-feldspar data in this study suggest that the melts forming A-type granite plutons in southwestern Fennoscandia and those forming pegmatites must have been derived from common source rocks in the continental crust that share their U-Th-Pb characteristics and age, i.e., a Palaeoproterozoic continental crust that has undergone internal fractionation in probably repeated events in the Mesoproterozoic [20]. The crustal protolith of the eastern areas (Subdomain C) had an on the average higher Th/U ratio than the western Subdomains A and B. Similar source characteristics is, however, neither evidence of a parent—derivative relationship between granites and pegmatites, nor of the absence of such a relationship. Crustal anatexis has played a major role in the evolution of the continental crust of SW Fennoscandia in late Mesoproterozoic and earliest Neoproterozoic time. A-type granitic magmatism and pegmatite formation are two related aspects of these processes.

## 6. Conclusions

The variations in Pb isotope compositions observed in K-feldspar from Sveconorwegian granitic pegmatites in important pegmatite fields in southern Norway show that the pegmatite-forming magmas and magmas that formed spatially associated A-type granitic plutons in the region have sampled melt sources that have similar U-Th-Pb and age properties. The pegmatite data thus add to the evidence from granitic plutons that the continental substrate below southern Norway is a continuation of the Palaeoproterozoic crust of the Svecofennian domain of Fennoscandia and the late Palaeoproterozoic Transscandinavian Igneous Belt.

The Pb isotope data from K-feldspar in granites and granite pegmatites cannot be used to support or reject petrogenetic models in which the two rock-types are connected to each other by parent magma—derivative melt relationships. They can, however, exclude petrogenetic models in which magmas form by direct anatexis of purely mantle-derived, mafic source rocks, since such processes would generate melts with less radiogenic Pb isotope compositions, in particular with lower $^{207}Pb/^{204}Pb$ than observed in the granite and pegmatite feldspars from southern Norway.

**Supplementary Materials:** The following supporting information can be downloaded at: https://www.mdpi.com/article/10.3390/min12070878/s1, Table S1: Radiogenic isotope data of K-feldspars from Sveconorwegian pegmatites and granites.

**Author Contributions:** Conceptualization, N.R.-S. and T.A.; methodology, T.A.; validation, N.R.-S., T.A. and A.M.; investigation, N.R.-S.; resources, A.M.; data curation, N.R.-S., T.A.; writing—original draft preparation, N.R.-S., T.A.; writing—review and editing, A.M.; visualization, N.R.-S.; supervision, T.A., A.M.; project administration, A.M. All authors have read and agreed to the published version of the manuscript.

**Funding:** This research received no external funding.

**Data Availability Statement:** The radiogenic isotope data of K-feldspars from Sveconorwegian pegmatites and granites (Supplementary Material Table S1) are publicly archived at the zenodo repository https://doi.org/10.5281/zenodo.6589459.svg.

**Acknowledgments:** We highly appreciate the support by Magnus Kristoffersen for the performance of the LA-ICPMS Pb isotope analyses. We thank the two anonymous reviewers for careful and constructive reviews and the editor of the MDPI Minerals for thoughtful editorial handling.

**Conflicts of Interest:** The authors declare no conflict of interest.

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
