# Peer review of "Lead Isotopes and the Sources of Granitic Magmas: The Sveconorwegian Granite and Pegmatite Province of Southern Norway"

_minerals, doi:10.3390/min12070878_

Round 1
Reviewer 1 Report
The paper by Rosing-Schow and co-workers presents Pb isotope data of K-feldspar from late Sveconorwegian granitic pegmatites and A-type, ferroan granites from southern Norway determined by LA-ICPMS to constrain the source of the felsic magmas.
The work is very interesting, and objectives and methods are clearly stated. I consider that the results are relevant and the work acceptable for publication.
However, there are many obvious typos and errors, so I recommend a careful reading of the manuscript to the authors. My only concern is that the spiking procedure reported in lines 279-282 should be better explained. How were the K-fds grains spiked with the Tl solution? Please add citations if the method is not novel.
Typos and minor corrections
General comment: please correct citations in the work. Put numbers of references in ascending order and indicate ranges where appropriate. For example (line 215) "77,83,84,78,85,86,81,87,88,37,72 " should be "37, 72, 77, 78, 81, 83-88"
Line 5: delete and
Line 12: analyzed by
Line 14: number 4 is wrong, please check it.
Line 15: check it (delete)
Line 50: have proved
Line 56: granitic melts have formed
Line 77: pegmatites cannot be related
Line 84: are relatively well known
Line 88: analyzed by
Line 130: suggest
Line 139: have
Lines 159-160: please rephrase
Figure 3: I don't find age 3 in the map
Line 239: are also found
Line 263: occurs
Figure 5: Please, indicate the subdomain in the map
Figure 10: Correct the label on the Y-axis of Fig. 10b.
Line 371: overlaps
Line 385: events of
Line 411: Rephrase, the 208Pb/204Pb ratio does not depend on 232Th/238U ratio. What depends on 232Th/238U ratio is the relation between 208Pb/204Pb and 206Pb/204Pb ratios.
Line 412: Please rephrase, the 208Pb/204Pb ratio does not depend on 232Th/238U ratio. What depends on 232Th/238U ratio is the relation between 208Pb/204Pb and 206Pb/204Pb ratios.
Line 413: Pb isotope composition
Line 416: Please rephrase. Which compositions?
Line 463: Figure 12: please correct: present-day as in line 436 or initial ?
Line 467: spreads
Line 482: had on the
Line 484: are, however,
Author Response
Response to Reviewer #1:
We highly appreciate the constructive and helpful corrections and comments by Reviewer #1, which considerably improved the quality of our manuscript. Please, find our response to the specific reviewer comments below. All language, spelling, grammar and format corrections suggested by the reviewer have been accepted and are not further commented upon. In addition, we did some minor improvements of Figures 2, 3 and 10, which are inserted in the uploaded revised manuscript version.
Reviewer comment: There are many obvious typos and errors, so I recommend a careful reading of the manuscript to the authors.
Response: We apologize for bad proof-reading before submission. Most of the issues pointed out are relevant, and we have either accepted the suggested changes or rephrased to make the intended meaning clear.
Reviewer comment: My only concern is that the spiking procedure reported in lines 279-282 should be better explained. How were the K-fds grains spiked with the Tl solution? Please add citations if the method is not novel.
Response: We recognize that the wording used was likely to cause misunderstanding, and have rephrased and added a reference to the method as it is used in our laboratory. Correcting for instrumental mass discrimination by aspiring a thallium solution with known isotopic composition into the transport line between ablation cell and plasma torch is, however, a well-established procedure in LA-ICPMS Pb isotope analysis.
Reviewer comment: please correct citations in the work. Put numbers of references in ascending order and indicate ranges where appropriate. For example (line 215) "77,83,84,78,85,86,81,87,88,37,72 " should be "37, 72, 77, 78, 81, 83-88"
Response: Done.
Reviewer 2 Report
This manuscript present high-quality Pb isotopic data of K-feldspar form the granites and pegmatites in southern Norway. Based on these data, the authors disproved the petrogenetic model that the pegmatites in the region formed by anatexis of mantle-derived mafic source. The manuscript is well written and presents reasonable interpretations. More importantly, it provides a reliable way that how to exclude a magma derived from purely mantle source by using Pb isotope. Collectively, I suggest that this manuscript can be accepted after minor revisions:
Some comments list below:
There are 101 articles have been cited in the manuscript, but only a few of them have been used in the “Discussion” section. In my opinion, some of interpretations (such as the lines 390-393; lines 479-481; lines 485-486) cannot be achieved based on the author’s present data, so I suggest the authors should add some references in the “Discussion” section.
Line 485-486: The authors should explain how the crustal anatexis played a major role in the continental evolution briefly. I think it can help explain why theses pegmatites cannot derived from a mantle source.
Line 498-500 The authors give an ambiguous conclusion about the relationship between the granites and pegmatites. I think this conclusion is not necessary in the manuscript if they cannot provide any useful information.
Author Response
Response to Reviewer #2
We highly appreciate the constructive and helpful corrections and comments by Reviewer #2, which considerably improved the quality of our manuscript. Please, find our response to the specific reviewer comments below. All language, spelling, grammar and format corrections suggested by the reviewer have been accepted and are not further commented upon. In addition, we did some minor improvements of Figures 2, 3 and 10, which are inserted in the uploaded revised manuscript version.
Reviewer comment: There are 101 articles have been cited in the manuscript, but only a few of them have been used in the “Discussion” section. In my opinion, some of interpretations (such as the lines 390-393; lines 479-481; lines 485-486) cannot be achieved based on the author’s present data, so I suggest the authors should add some references in the “Discussion” section.
Response: We realize that the reference list is long. This has been necessary because the rocks we are dealing with occur in a complex, poly-orogenic geological setting where there is (1): No published overview paper that covers the region in the necessary detail, and (2): Some important features of crustal evolution history remain controversial to this day. The reviewer is correct in pointing out that the discussion is short on references. The missing information can be found in papers already referred to in earlier sections of the paper, and we have now invoked these where relevant. This, for example, applies to the specific comments to lines 485-486.
Reviewer comment: Line 498-500 The authors give an ambiguous conclusion about the relationship between the granites and pegmatites. I think this conclusion is not necessary in the manuscript if they cannot provide any useful information. Line 485-486: The authors should explain how the crustal anatexis played a major role in the continental evolution briefly. I think it can help explain why theses pegmatites cannot derived from a mantle source.
Response: There are, in general, two possible mechanisms to generate the highly evolved melts forming granite pegmatites: Magmatic differentiation of some parent magma, or direct formation of small-volume partial melts of suitable composition. Whereas lead isotope data can give highly relevant information on the ultimate source of material in the pegmatites, they cannot be used to distinguish between the two petrogenetic mechanisms. The ambiguity is thus real, and we want to emphasise the point. We extended the discussion to be more clearly.